# Local-Global Based High-Resolution Spatial-Spectral Representation Network for Pansharpening

**Wei Huang \*, Ming Ju, Zhuobing Zhao, Qinggang Wu and Erlin Tian**

College of Computer and Communication Engineering, Zhengzhou University of Light Industry, Zhengzhou 450002, China; juming@email.zzuli.edu.cn (M.J.); zhaozhuobing@email.zzuli.edu.cn (Z.Z.); 2013009@zzuli.edu.cn (Q.W.); 2004022@zzuli.edu.cn (E.T.)

\* Correspondence: hnhw235@zzuli.edu.cn; Tel.: +86-182-3714-0566

**Abstract:** Due to the inability of convolutional neural networks to effectively obtain long-range information, a transformer was recently introduced into the field of pansharpening to obtain global dependencies. However, a transformer does not pay enough attention to the information of channel dimensions. To solve this problem, a local-global-based high-resolution spatial-spectral representation network (LG-HSSRN) is proposed to fully fuse local and global spatial-spectral information at different scales. In this paper, a multi-scale feature fusion (MSFF) architecture is designed to obtain the scale information of remote sensing images. Meanwhile, in order to learn spatial texture information and spectral information effectively, a local-global feature extraction (LGFE) module is proposed to capture the local and global dependencies in the source images from a spatial-spectral perspective. In addition, a multi-scale contextual aggregation (MSCA) module is proposed to weave hierarchical information with high representational power. The results of three satellite datasets show that the proposed method exhibits superior performance in terms of both spatial and spectral preservation compared to other methods.

**Keywords:** pansharpening; transformer; texture; high-resolution; depthwise separable convolution; contextual aggregation

## 1. Introduction

In recent years, with the development of remote sensing technology, more and more sensors have been applied to Earth observation, which provides users with a wealth of remote sensing information. Due to technical limitation, information from a single sensor cannot fully reflect all the characteristics of the target area. Therefore, a satellite is usually fitted with multiple sensors to acquire remote sensing images containing different information separately, and at the same time, fusion techniques are used to construct a single remote sensing image that fully reflects the features of the ground [1]. Pansharpening is a popular remote sensing image fusion technique, which focuses on fusing high spatial resolution panchromatic (PAN) image and low-resolution multispectral (LRMS) images to generate high spatial resolution multispectral (HRMS) images. The fused results are widely used in target detection [2], land cover classification [3], geological exploration, and other fields.

Nowadays, a large number of excellent pansharpening methods have emerged. These methods can be broadly classified into four categories based on component replacement (CS), multi-resolution analysis (MRA), variational optimization, and deep learning.

The main principle of CS-based methods is to project the LRMS images into another space by domain transformation and replace all spatial information components using the PAN image and then invert them to obtain the final fused images. The common methods in this category are principal component analysis [4], partial replacement adaptive CS (PRACS) [5], the band-related spatial detail scheme [6], intensity-hue-saturation [7],

Gram-Schmidt (GS) [8], etc. These CS-based methods are simple in structure, easy to implement, and can effectively enhance the spatial information of the generated results, but they ignore the differences between PAN and MS images, which can easily lead to spectral distortion.

The MRA-based method is as simple and easy to implement as CS-based methods. MRA is mainly based on obtaining the corresponding spatial information from the PAN image and injecting this information into the low-resolution MS images. Wavelet [9], high-pass filters (HPF) [10], generalized Laplacian pyramid (GLP) [11], smoothing filter-based intensity modulation [12], and nonsubsampled contourlet [13] are among the MRA methods. These methods are able to obtain spectrally well-preserved results, but at the cost of spatial distortion. In view of the limitations of CS and MRA methods, some hybrid methods have been proposed, such as Revisited AWLP [14], a pansharpening method using guided filter [15], etc. These methods are mainly designed to obtain a good spatial enhancement while maintaining spectral consistency. However, it is experimentally proven that these methods still produce some spectral and spatial distortions. Both the CS and MRA types of methods can be classified as conventional.

Variational optimization-based methods consider pansharpening as a problem to be solved optimally, and seek the optimal balance between maintaining spectral consistency and improving spatial quality in the fusion process of PAN and MS images by designing specific models. Such methods mainly include Bayesian posterior probability [16], sparse reconstruction-based fusion methods [17], P+XS [18], variational pansharpening with local gradient constraints [19], etc. These methods mainly rely on a large amount of prior knowledge to constrain the constructed spectral-spatial solution models, which can successfully reconstruct HRMS images with superior spectral and spatial information. However, a priori information used in the models may be invalid for data in complex scenarios, which leads to limitations in the application of variational optimization-based methods. Moreover, the construction of the whole model relies excessively on a priori knowledge, and the architecture design is complicated and computationally intensive.

With the development of deep learning, many advanced deep learning techniques applied to the pansharpening field emerged. Benefiting from the powerful feature extraction capability of deep learning, the fusion performance of pansharpening algorithms has been greatly improved. Huang et al. [20] proposed a sparse noise-reducing self-coding pansharpening method to obtain the relationship between high-resolution and low-resolution images using deep neural networks, and obtained a relatively good fusion effect. Masi et al. [21] were influenced by the SRCNN model [22] in the field of super-resolution reconstruction (SR) and applied convolutional neural networks (CNNs) to the field of pansharpening (PNN) for the first time, which greatly improved the performance of the algorithm compared with traditional methods. Wei et al. [23], on the other hand, incorporated the idea of residuals into the fusion network and proposed the deep residual generalized sharpness neural network, which effectively alleviates the information loss problem during feature extraction. Fu et al. [24] developed PanNet, which designed the corresponding network framework for the two specific objectives of spectral preservation and spatial enhancement, and obtained better fusion results. Yuan et al. [25] proposed a multi-scale multi-depth neural network (MSDCNN), using convolutional kernels of different sizes to extract multi-scale features and enhance the performance of the pansharpening algorithm from the perspective of widening the network. Ma et al. [26] designed a generative adversarial network (Pan-GAN) containing spectral and spatial dual discriminators, which made full use of the unsupervised characteristics of GAN networks and achieved more robust results. Liu et al. [27] propose a novel pansharpening architecture that constructs a deep-shallow network to extract multi-level spatial information from PAN images in the high-frequency domain, and uses a spectral injection network to map the spatial information into the various bands of MS images, effectively improving the overall fusion quality. Wei et al. [28] then developed a two-stream fusion network based on asymmetric convolution, following the architecture of Liu et al. [27]. Wang et al. [29] proposed

an SSConv that converts spectral information to the spatial domain for upsampling to reduce the artifacts associated with normal upsampling, and a novel U-shaped network to fuse information from multiple source images. Deng et al. [30] combined the traditional CS and MRA methods with CNN and proposed FusionNet, which achieved more robust results with a simple architecture.

Recently, since CNNs cannot effectively acquire global information between images, the transformer architecture has been introduced to the field of computer vision in order to learn the non-local dependencies of images, and the transformer can effectively acquire non-local information through a multi-head global attention mechanism. In the field of pansharpening, Zhou et al. [31] first applied the transformer to image fusion and designed a pansharpening transformer module that can effectively integrate complementary information between different images and accurately acquire long-range spatial information. Nithin et al. [32] proposed a transformer-based self-attentive network (Pansformer), which utilizes a non-overlapping multi-patch attention mechanism to obtain details of non-local information and is able to produce a higher quality HRMS images. Although the abovementioned transformer-based models alleviate the shortcomings of CNNs and are able to effectively capture long-range dependencies, they ignore the scale effects of remote sensing images. Meanwhile, the transformer has high complexity for spatial quadratic computation on high-resolution images and ignores the channel dimensionality adaptation of the images.

To address the above problems, we propose a local-global based high-resolution spatial-spectral representation network (LG-HSSRN). First, a local-global feature extraction (LGFE) module using a multiscale residual block convolution component, texture-transformer and Multi-Dconv transformer to obtain global texture information and global context information across channels. Then a multi-scale context aggregation (MSCA) module is used to map the corresponding information at different scales to the high-resolution level to obtain semantic information at different scales. Finally, a multi-stream feature fusion (MSFF) module is performed on the different information to reconstruct the final HRMS images.

The main contributions of this paper are as follows:

- The LG-HSSRN that can effectively obtain local and global dependencies is proposed. At the same time, the information at different scales is mapped to the output scale layer, which maintains a high-resolution representation and can obtain better contextual information.

- Considering the complementary characteristics of the information contained in PAN and MS images, the LGFE module is designed, which can effectively obtain local and non-local information from images. Among others, we designed a texture-transformer to extract long-range texture details and cross-feature spatial dependencies from a spatial perspective. A Multi-Dconv transformer module is designed to learn contextual image information across channels using a self-attentive mechanism and is able to aggregate local and non-local pixel interactions.

- The MSCA module is proposed to map all low-level features and mid-level feature information to the high level. The final feature fusion is completed with a high-resolution feature representation while fully obtaining the hierarchical information.

The remainder of this paper is organized as follows. Section 2 describes the details of the proposed LG-HSSRN. The experimental results and analysis of different datasets are given in Section 3 for validating the effectiveness of the LG-HSSRN algorithm. The overall experimental discussion is given in Section 4. Conclusions are given in Section 5.

## 2. Proposed Method

The overall framework of the proposed method is shown in Figure 1 and consists of three main modules: the LGFE module, MSCA module and MSFF module. The LGFE module is composed of two components: a global feature extraction module consisting of

a texture-transformer for learning global texture information and a Multi-Dconv transformer for acquiring the spatial information from each image band across channels, and a local feature extraction module consisting of multi-scale residual blocks.

First, the input PAN and MS images with three different scales of image pairs are obtained by upsampling and downsampling operations. Then, the three different scale image pairs are input to the LGFE module of the corresponding layers, and each layer has three outputs with different information, including long-range texture information, cross-channel global context information, and local multi-scale information. Moreover, we map the different feature maps of three different scales to the high-resolution level through the MSCA module with corresponding kinds of outputs to obtain three different feature maps containing multi-scale contextual information. Finally, the three different types of high-resolution feature information are fused by the MSFF module to reconstruct the final HRMS images.

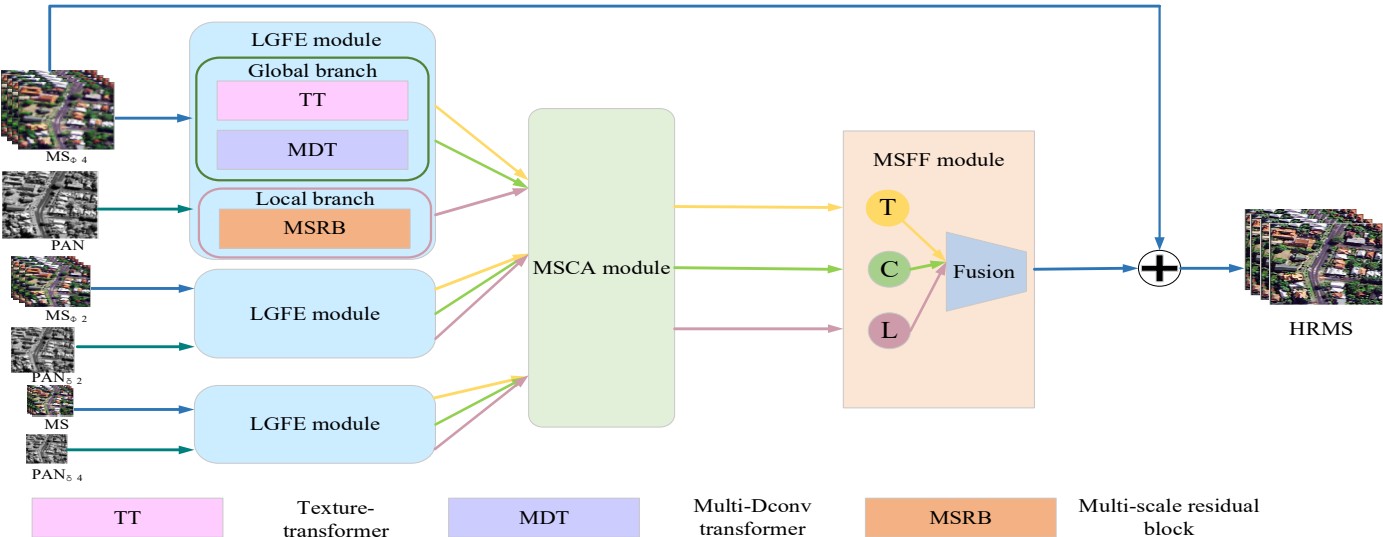

**Figure 1.** The overall framework diagram of the LG-HSSRN. The arrows in different colors indicate the data flow of different types of features.

*2.1. LGFE Module*

2.1.1. Global Feature Exaction Module

Due to the similarity of texture information between PAN and MS images, extracting the corresponding spatial information from the PAN image alone or combining the two input processes cannot fully utilize the spatial information in the PAN image, and it is easy to cause spatial and spectral distortion. Therefore, we design a texture-transformer module for learning global spatial texture information based on the different information characteristics of PAN and MS images. In addition, we design the Multi-Dconv transformer module for acquiring global information from the channel dimension, which is used to acquire the contextual information of each band of the image across channels.

(1)    Texture-transformer Module

Since the texture-transformer is mainly used to learn similar texture information from the two images [33] to be used to better obtain global spatial information, we use different inputs as shown in Figure 2. Specifically, we first segment the PAN and MS images into non-overlapping patches of the same size, and encode their positions to form the corresponding images patches sequence $[p_1, \cdots\cdots, p_n]$ with $[MS_1, \cdots\cdots, MS_n]$. The image patches are used as the texture features to be input, meanwhile, the features are extracted using $3\times3$ convolution and projected into the three components necessary for the

corresponding transformer, namely: $Q$ (query), $K$ (key), and $V$ (value), respectively. The mathematical expressions are as follows:

$$Q = Conv([MS_1, \cdots\cdots, MS_n]) \tag{1}$$

$$K = Conv([p_1, \cdots\cdots, p_n]) \tag{2}$$

$$V = Conv([p_1, \cdots\cdots, p_n]) \tag{3}$$

where $Conv(\cdot)$ denotes the $3 \times 3$ convolution operation. Subsequently, we use $Q$ and $K$ for estimating the similarity to generate the association matrix, and multiply the obtained association matrix with the PAN image features, i.e., $V$, to obtain the final weighted feature map. In addition, we also introduce a jump connection to prevent information loss. Finally, all the generated patch feature maps are stitched together again by encoding position to obtain the final texture feature map as follows:

$$S_i = soft\max(\frac{Q_i^T K_i}{\sqrt{d_k}}) \tag{4}$$

$$Texturemap_i = S_i \cdot V_i + V_i \tag{5}$$

$$Texturemap = Concat([Texturemap_1, \cdots\cdots, Texturemap_n]) \tag{6}$$

where $S_i$ is the weight matrix of the image patches corresponding to encoding position $i$, $\sqrt{d_k}$ is the score normalization used in the transformer for gradient stabilization, and $soft\max$ is the corresponding nonlinear activation function, which is used to normalize the feature weights. $Texturemap_i$ corresponds to the texture feature of the image patches with encoding position $i$. $Concat$ is the stitching together of the generated texture feature image patches according to the encoding position. $Texturemap$ is the final output texture feature map.

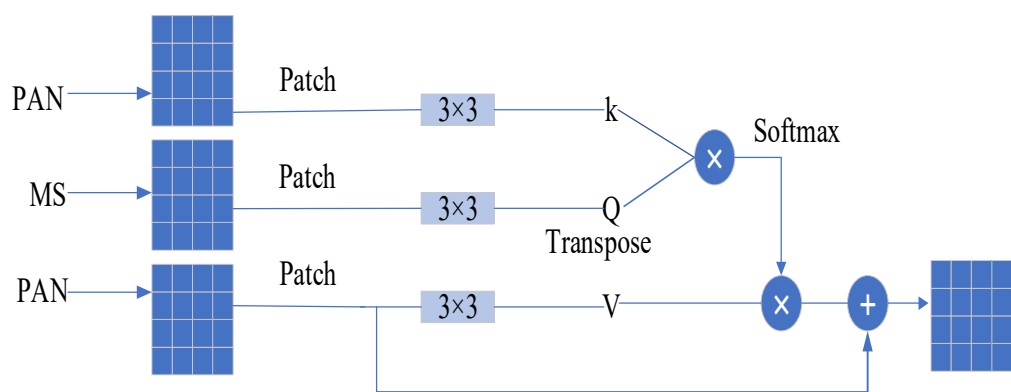

**Figure 2.** The specific construction of texture-transformer module.

In a word, we obtain the image patches of different parts by spatially fragmenting the overall image, and perform a similarity calculation and self-feature enhancement of these image patches to obtain the attention-enhanced maps of different parts. Finally, the attentional enhancement maps of different locations are stitched together to obtain the global attentional enhancement map.

(2)　Multi-Dconv Transformer Module

　　　Since the transformer mostly concentrates on the spatial dimension to learn long-range dependencies, it does not consider learning contextual relationships from the channel dimension. Meanwhile, the image binning operation is able to obtain local contextual relations cumulatively [34], but it is computationally intensive for high-resolution images and does not achieve true global contextual aggregation. Therefore, we embed deep separable convolution into the self-attentive mechanism and use the Multi-Dconv transformer module to alleviate these problems by obtaining the corresponding spatial context information from the channel dimension.

　　　As shown in Figure 3, in order to fully acquire the features of the PAN and MS images, we choose to jointly input both into the module. First, we use point convolution of $1\times1$ to obtain contextual information between different channels at the pixel level, and subsequently use $3\times3$ depth convolution to obtain spatial context along the channel direction. Subsequently, we map the contextual information learned from the depth-separable convolution to $Q$, $K$ and $V$ as follows:

$$X = Concat(M, P) \tag{7}$$

$$Q = W_P W_d X \tag{8}$$

$$K = W_P W_d X \tag{9}$$

$$V = W_P W_d X \tag{10}$$

where $M$ and $P$ correspond to the input MS and PAN images, respectively, $X$ represents the union of the two images, and $W_P$ and $W_d$ represent the $1\times1$ point convolution and $3\times3$ depth convolution operations, respectively. Finally, the operations of Equations (4) and (5) are repeated to obtain the final cross-channel contextual information as follows:

$$Channel\, attention\, map = V \cdot soft\max(\frac{Q^T K}{\sqrt{d_k}}) \tag{11}$$

　　　It is worth noting that we do not perform a segmentation patch operation on the input, but use the features obtained from the image of each channel as the head in a multi-headed self-attentive mechanism, which enables a good interaction of local and non-local pixels and is sufficient to obtain the rich information in the spectral images. This is explained in detail by using point and depth convolution for spatial and channel information extraction of local information, and inputting the acquired information into the self-attentive mechanism to obtain the long-range dependencies of the image, thus enabling the interaction between local and non-local pixels.

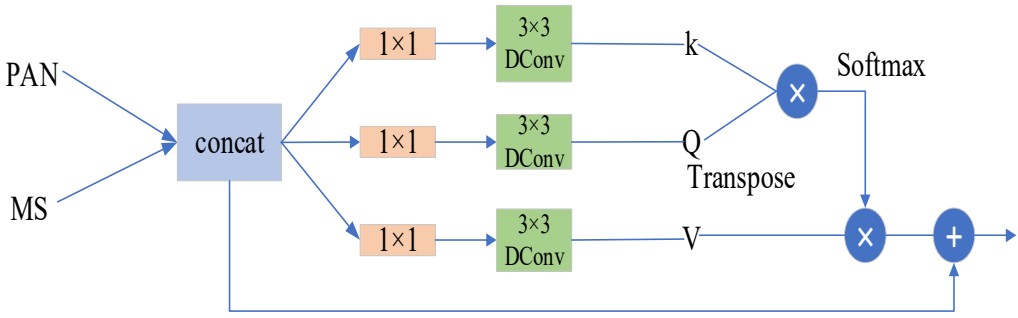

**Figure 3.** The specific construction of the Multi-Dconv transformer module.

The common multi-head attention mechanism is used to obtain a non-local attention map by point multiply in the spatial dimension. whose computational complexity increases with the spatial resolution of the image (computational complexity is the square of the spatial resolution). Our Multi-Dconv transformer module is used to generate the input non-local attention map across channel dimensions, and its computational complexity is linear and much smaller than the spatial quadratic computational complexity of the multi-head attention mechanism. This overcomes the high spatial computational complexity associated with the normal multi-head attention mechanism while capturing the long-range dependencies across channel dimensions.

### 2.1.2. Local Feature Exaction Module

For the local feature extraction branch, as shown in Figure 4, we simply use a multi-scale residual block (MSRB) composed of convolutions of different sizes to extract the multi-scale features of the input image and superimpose them to improve the representational power of local features, while using jump connections to complement them with information and mitigate information loss during convolution.

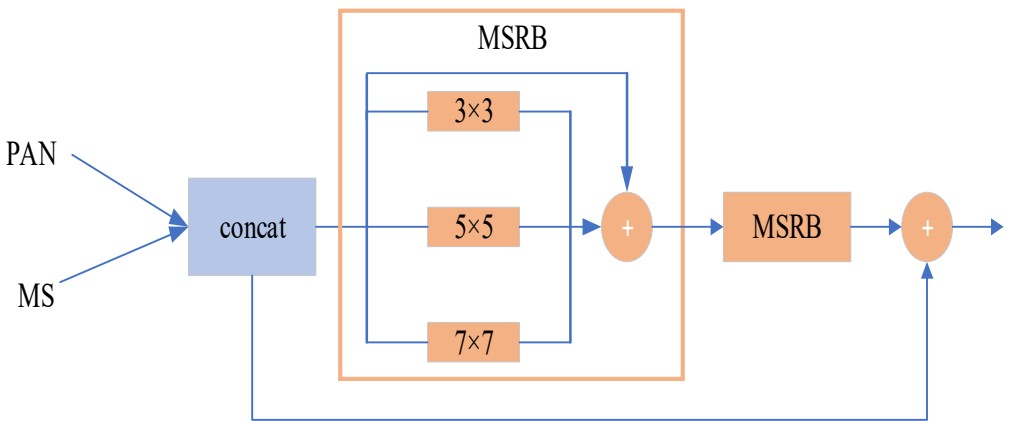

**Figure 4.** The construction of local feature extraction branch.

### 2.2. MSCA Module

Since pansharpening as the task of transforming low resolution to high resolution, the final output is a high-resolution MS images. Remote sensing images also have scale effects, i.e., different scales contain different feature information. Therefore, for the fusion process of multi-scale multi-features, we focus on maintaining the high-resolution feature representation. Inspired by HRNet [35], we design the MSCA module to aggregate the same type of features at different scales.

The overall construction of the MSCA is shown in Figure 5a, where we aggregate different scales of the same type of features to the highest resolution level by the up-sampling operation, use $1 \times 1$ convolution to mix the semantic and feature details of the different scales, and finally obtain three different types of high variability feature maps containing rich contextual information. In this case, the high-resolution mapping of a single kind of information is shown in Figure 5b, where three different scales of the same kind of features are mapped to the highest resolution size by $1X, 2X, 4X$ upsampling operations for a superposition operation, followed by information aggregation using convolution. The specific operation flow of MSCA is shown in the following:

$$\begin{cases} T = Conv(T_{1x} + T_{2x} + T_{4x}) \\ C = Conv(C_{1x} + C_{2x} + C_{4x}) \\ L = Conv(L_{1x} + L_{2x} + L_{4x}) \end{cases} \tag{12}$$

where $T$, $C$, and $L$ represent the final aggregated texture features, cross-channel context information, and local features, respectively. *Conv* represents the $1\times1$ convolution operation, and the subscripts $1X, 2X, 4X$ are used to label the features with different resolutions.

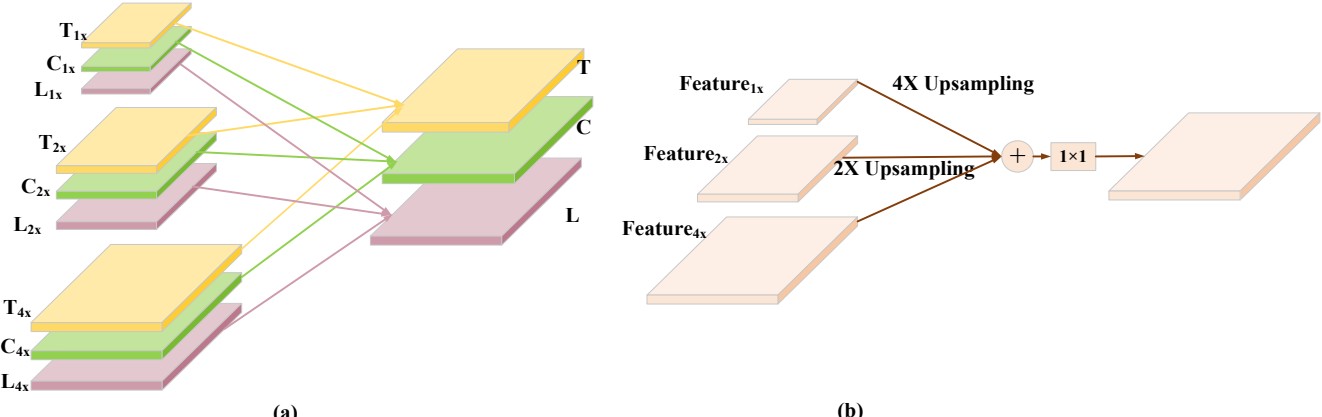

**(a)**                  **(b)**

**Figure 5.** The architecture and details of the MSCA Module. (**a**) The overall architecture diagram of the MSCA module, (**b**) The flow chart of homogeneous feature aggregation.

### 2.3. MSFF Module

We designed the MSFF module for final feature fusion and image reconstruction, with an architecture as shown in Figure 6. We first fuse the global texture information with the cross-channel global context information and use a convolutional block for global information integration. Next, the global information is fused with the local information. Finally, a convolution block is used again for the final information integration to output the final reconstructed images.

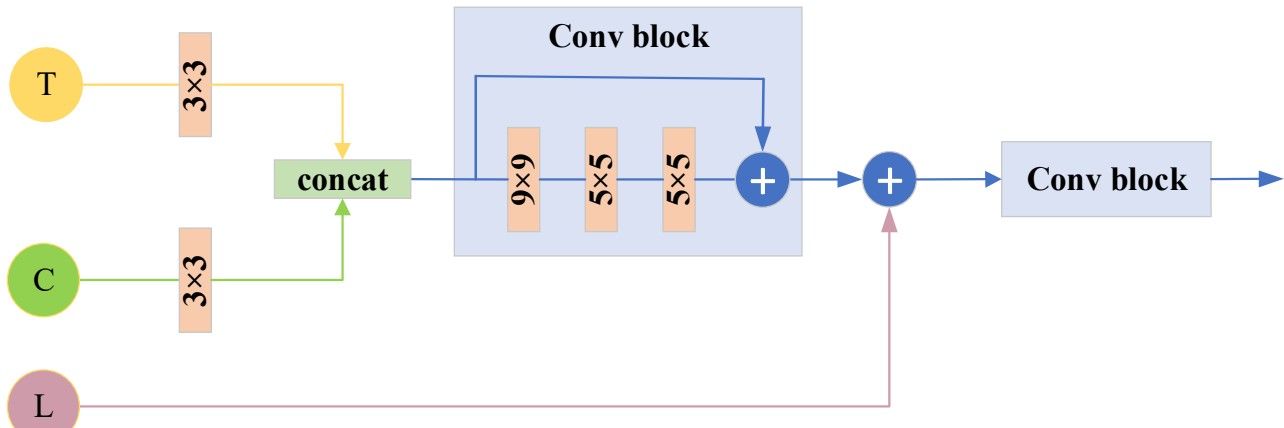

**Figure 6.** The architecture of MSFF module.

### 2.4. Loss Function

The root mean square error is used as a loss function that constrains the whole network, optimizing the network by minimizing the error between the ground truth (GT) images and the fusion result. The loss function equation is as follows:

$$Loss = \frac{1}{N}\sum_{i=1}^{N}\left\|GT_i - F\left(M_i, P_i\right)\right\|_F^2 \tag{13}$$

where $GT_i$ is the $i$-th training sample in the ground truth images, $M_i$ represents the $i$-th training sample in the low-resolution MS images, $P_i$ represents the $i$-th training sample in the PAN image, and $N$ is the number of training samples randomly selected from the training set in one iteration. $Loss$ is the loss function minimized during the training process.

## 3. Experiments and Results

In this section, we introduce the experimental dataset, comparison method, and experimental setup. We show the performance of the method proposed in this paper and other methods in different datasets. Finally, we validate the performance of each module in the proposed model.

### 3.1. Experimental Data

In this experiment, PAN and MS images captured by three satellites, GaoFen-2, WorldView-2 and QuickBird, were used to form three different datasets. The MS images from these satellites all contain four bands: red, green, blue, and NIR. In addition, we selected the red, green, blue, and NIR bands from the MS images of the WorldView-2 satellite to form a new 4-band MS images. Table 1 shows the spatial resolution of these satellite datasets.

**Table 1.** The spatial resolution of each satellite image.

| Satellite | Band | Resolution (m) |
|-----------|------|----------------|
| GaoFen-2 | MS | 4 |
| | PAN | 1 |
| WorldView-2 | MS | 1.6 |
| | PAN | 0.4 |
| QuickBird | MS | 2.4 |
| | PAN | 0.6 |

To facilitate uniform training and testing, we divided the datasets of the three satellites into uniform image sizes. Based on Wald's protocol [36], the artificial datasets instead of the real datasets were divided into the training and testing datasets. In this case, the ratio of training to validation in the training set was divided 4:1. In the training set, we patched the original MS images as ground truth (GT) images. All the training set MS/PAN image pairs were 16/64 in size, simulated experimental image pairs were 128/512 in size, and real experimental image pairs were 256/1024 in size. The details are shown in Table 2.

**Table 2.** The specific settings for simulation and real datasets.

| Dataset | Kind | Satellite | Size | Number |
|---------|------|-----------|------|--------|
| Training dataset | Simulated experiment | GaoFen-2 | 16 × 16, MS<br>64 × 64, PAN | Training, 6812<br>Validation, 1703 |
| | | WorldView-2 | 16 × 16<br>64 × 64 | Training, 1819<br>Validation, 452 |
| | | QuickBird | 16 × 16<br>64 × 64 | Training, 2779<br>Validation, 694 |
| Testing dataset | Simulated experiment | GaoFen-2 | 128 × 128, MS<br>512 × 512, PAN | 52 |
| | | WorldView-2 | 128 × 128<br>512 × 512 | 33 |

| | | | |
|---|---|---|---|
| | QuickBird | 128 × 128<br>512 × 512 | 33 |
| Real experiment | GaoFen-2 | 256 × 256<br>1024 × 1024 | 100 |
| | WorldView-2 | 256 × 256<br>1024 × 1024 | 100 |
| | QuickBird | 256 × 256<br>1024 × 1024 | 100 |

### 3.2. Comparison Methods

In order to verify the effectiveness of our proposed approach, we selected eight different methods as our comparison experiments and conducted simulations and real experiments in the three datasets described above. Among them, the main deep learning-based methods are the PNN [22], MSDCNN [25], MUCNN [29], and Pansformer [32], which are four excellent methods. Due to the large time overhead, we did not cover the variational optimization-based approach. The traditional CS and MRA algorithms mainly include GS [8], PRACS [5], Wavelet [9] and GLP with MTF-matched filter (MTF-GLP) [37] methods. Moreover, to ensure the fairness of the experiments, all the traditional methods were implemented on MATLAB 2018b. All deep learning-based algorithms were implemented on the PC side using NVIDIA GeForce RTX 3060 GPUs in the PyTorch framework.

The training parameters of the deep learning-based model are shown in Table 3. We set the training iterations to 1200 because the number of iterations should be chosen to fit the current amount of data, and too many iterations will inevitably reduce the efficiency of the algorithm. The normal pytorch-based fusion framework was set to 1000 iterations; we have also tried 1500 and 2000 iterations, but found that 1200 is the most appropriate, which is the parameter setting chosen by most PyTorch-based fusion algorithms. We set the training batch size to 16, as the transformer-based algorithm requires a certain amount of computational resources, and too large or too small a batch size will affect the computational efficiency and the final fusion results. We chose to use Adma as the optimizer for all deep learning-based fusion networks because, compared to SGD, the dominant optimizer, Adma is able to avoid local optima and to design independent adaptive learning rates for different parameters by calculating first- and second-order moment estimates of the gradients with little or no fine-tuning. Therefore, the typical values of Adma's parameters, i.e., 0.001 and (0.9,0.999), are used for both the learning rate and the decay factor, which are also the parameter settings of most fusion models. During the training of the model, we performed validation every 50 batches and saved the best parameter model in the validation set every 100 batches for simulation and real testing.

**Table 3.** Parameter setting of the training model.

| Iterations | Batch Size | Optimizer | Learning Rate | Decay Rate |
|---|---|---|---|---|
| 1200 | 16 | Adam | 0.001 | (0.9, 0.999) |

### 3.3. Evaluation Metrics

In order to comprehensively evaluate the experimental results, we evaluated the fusion effect from both subjective and objective perspectives. The subjective evaluation mainly relies for its determination on the observation of the pseudo-color maps of the generated results. The objective perspective mainly relies on the reference indicators used to evaluate the simulated experimental results and the non-reference indicators used to evaluate the real experimental results. The reference indicators include the relative global synthesis error (ERGAS) [38], the spectral angle mapper (SAM) [39], the correlation coefficient (CC) [10], a universal image quality index (UIQI) [40] and its extended index *Q4* [41]. Among them, the SAM is the most commonly used spectral index, which represents

the angle between the reference vector and the processing vector of a pixel in the image spectral feature space. ERGAS calculates the quality of the fused image as the normalized mean error of the fused image band, which ranges from zero to infinity, with lower values indicating a higher degree of similarity between the two images. The CC reflects the geometric distortion of the image. UIQI and $Q4$ are the universal image quality indexes for each and all bands between the fused image and the reference image, with values ranging from 0 to 1. A value of 1 indicates that the two images are perfectly similar.

The reference-free quality index (QNR) [42] consists of two main components: the spectral quality index $D_\lambda$ to assess the spectral quality of the image and the spatial quality index $D_s$ to measure the structural performance of the pansharpening results; the closer the values of both indexes to 0, the better the corresponding spectral and spatial quality. The QNR evaluates the overall pansharpening performance of the generated images.

*3.4. Simulation Experiment Results and Analysis*

In this section, we perform simulation experiments on the WorldView-2, GaoFen-2 and QuickBird datasets for 9 different methods and give the corresponding metric results and visualization results, respectively. In addition, to facilitate the observation of the information in the visualization results, we not only place the detailed local zoomed-in maps of the corresponding methods, but also show the mean square error maps based on the GT images.

Table 4 shows the objective evaluation metrics of all methods on the WorldView-2 dataset, where we bolded the best value of each metric. It is found that the deep learning-based methods outperform traditional methods in most of the metrics. Among them, the PNN and MSDCNN, have close values of the finger table on spectral and spatial conservation. While the MUCNN and Pansformer perform better than the PNN and MSDCNN in terms of numerical values and higher overall image quality, the LG-HSSRN is superior to the other methods in all metrics.

**Table 4.** Quantitative evaluation metrics for the WorldView-2 simulation dataset.

| Method | SAM | ERGAS | CC | UIQI | Q4 |
|---|---|---|---|---|---|
| **Reference** | **0** | **0** | **1** | **1** | **1** |
| GS | 4.7293 | 6.4284 | 0.8385 | 0.7770 | 0.7626 |
| PRACS | 5.965 | 3.7119 | 0.9031 | 0.8216 | 0.8042 |
| MTF-GLP | 3.2137 | 6.7711 | 0.8887 | 0.8649 | 0.8351 |
| Wavelet | 2.6919 | 6.8493 | 0.8749 | 0.8057 | 0.7843 |
| PNN | 3.9862 | 5.6703 | 0.9661 | 0.8913 | 0.8805 |
| MSDCNN | 3.9734 | 5.6449 | 0.9776 | 0.9257 | 0.9159 |
| MUCNN | 3.2673 | 4.2673 | 0.9838 | 0.9482 | 0.9375 |
| Pansformer | 2.7412 | 3.8938 | 0.9826 | 0.9590 | 0.9453 |
| LG-HSSRN | **1.5060** | **2.9222** | **0.9882** | **0.9731** | **0.9645** |

Figures 7 and 8 show the local zoom and difference maps, respectively, of the fusion results of the various methods on the WorldView-2 dataset and it can be seen that MTF-GLP and Wavelet show significant spatial distortion and the overall image appears blurred. The overall architecture of the deep learning-based method is clearer, and the local zoomed-in parts do not show obvious contour blurring. By comparing the difference maps of the deep learning methods, it can be seen that the errors of the PNN and MSDCNN with GT are larger. Through the location of the dock in the image, it is found that the error brightness of the MUCNN and Pansformer is weaker than in the previous cases, but the results generated by the LG-HSSRN are closer to the GT images.

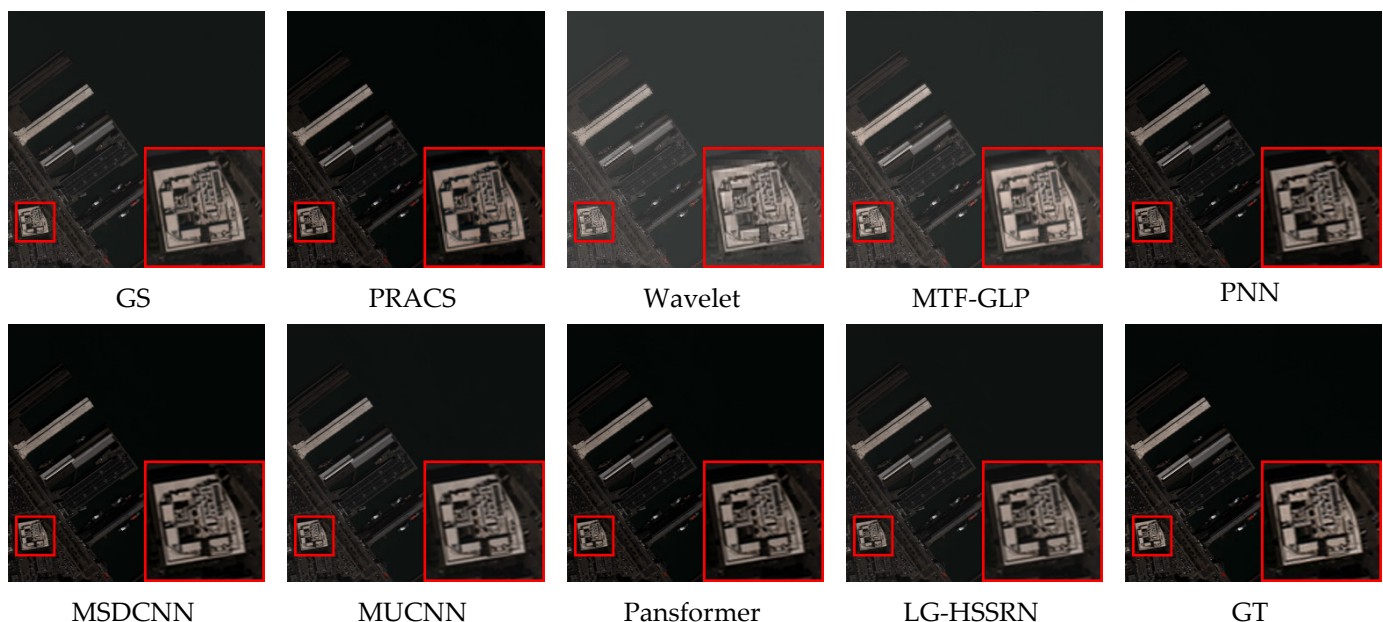

**Figure 7.** Comparison of pseudo-color maps of various methods on the WorldView-2 simulated dataset.

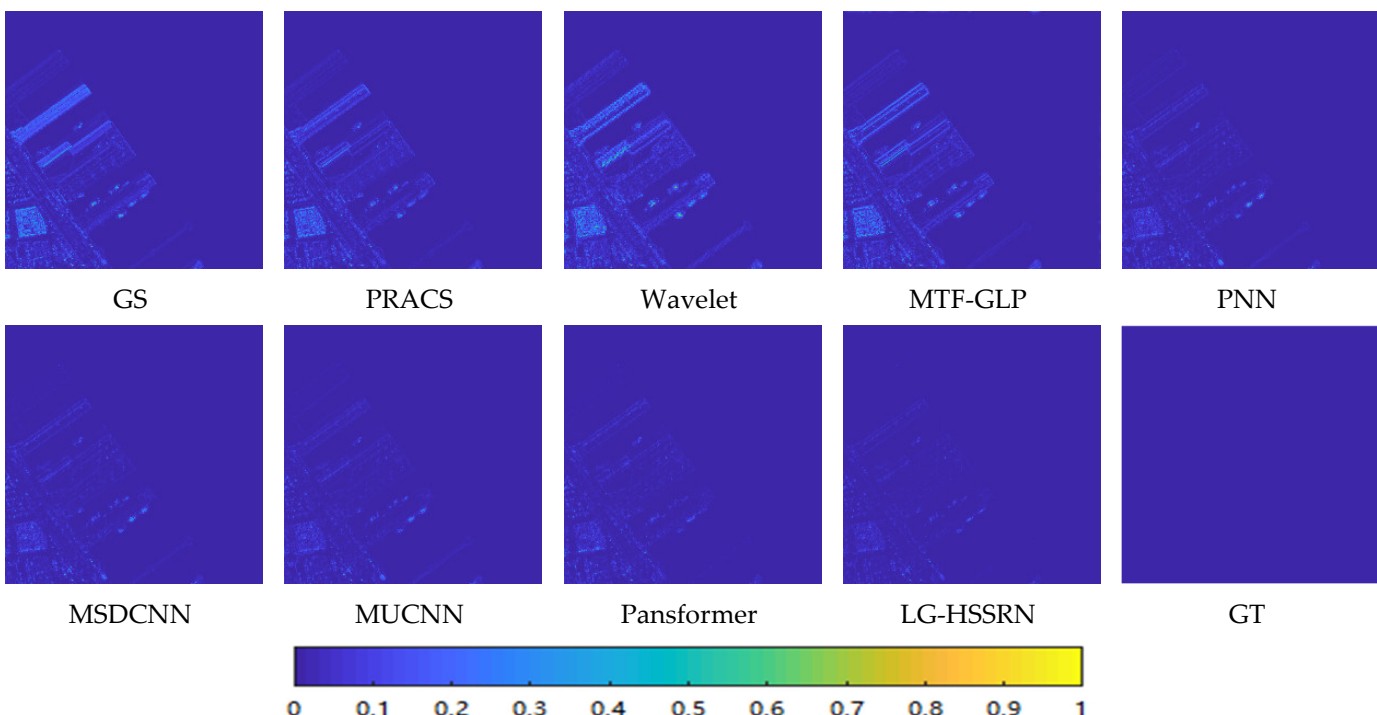

**Figure 8.** Comparison of mean square error maps of various methods on the WorldView-2 simulated dataset.

Figures 9 and 10 show the pseudo-color maps and the difference maps, respectively, of different methods on the GaoFen-2 dataset. It is clearly seen that GS and PRACS have obvious color blurring, and the spatial and color distortion of buildings in the images appear obvious. The spatial structure of the overall image of MTF-GLP and Wavelet is destroyed, and obvious spatial blurring appears. The PNN mitigates the spatial distortion, but obvious spectral distortion appears, and the red roof color degree is too light. The MSDCNN, MUCNN and Pansformer are closer in spatial structure and spectral retention, but still show some spatial blurring phenomenon.

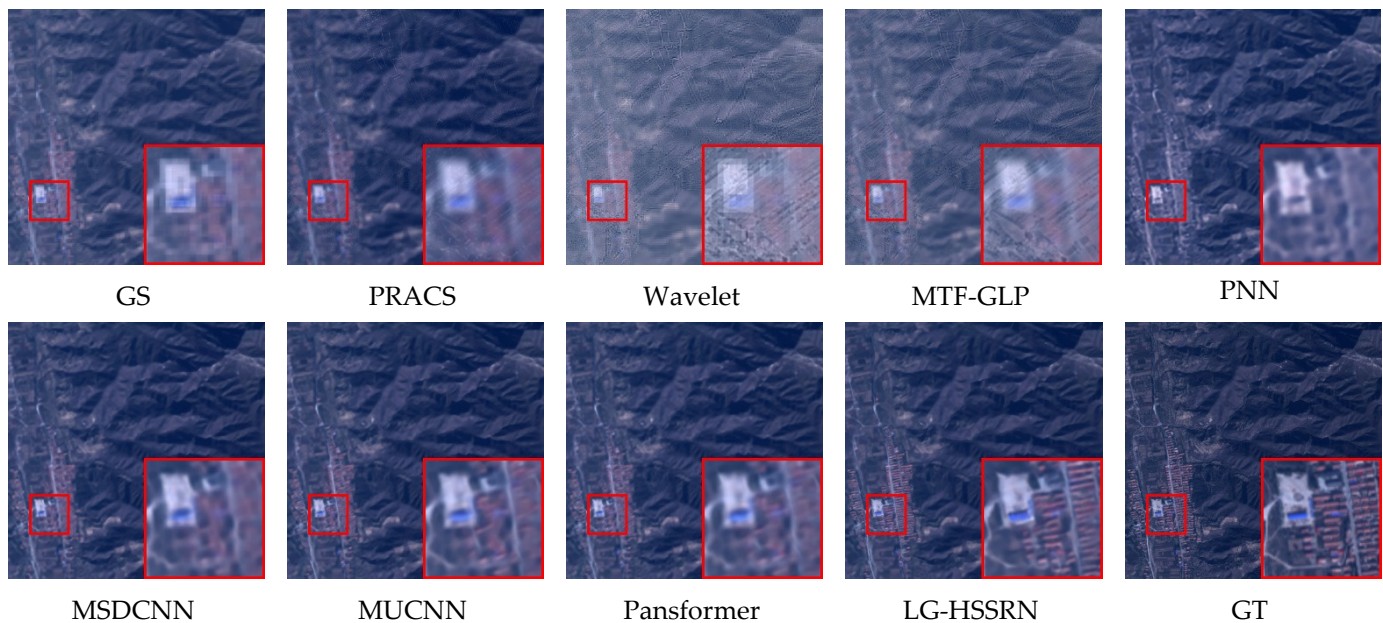

**Figure 9.** Comparison of pseudo-color maps of various methods on the GaoFen-2 simulated dataset.

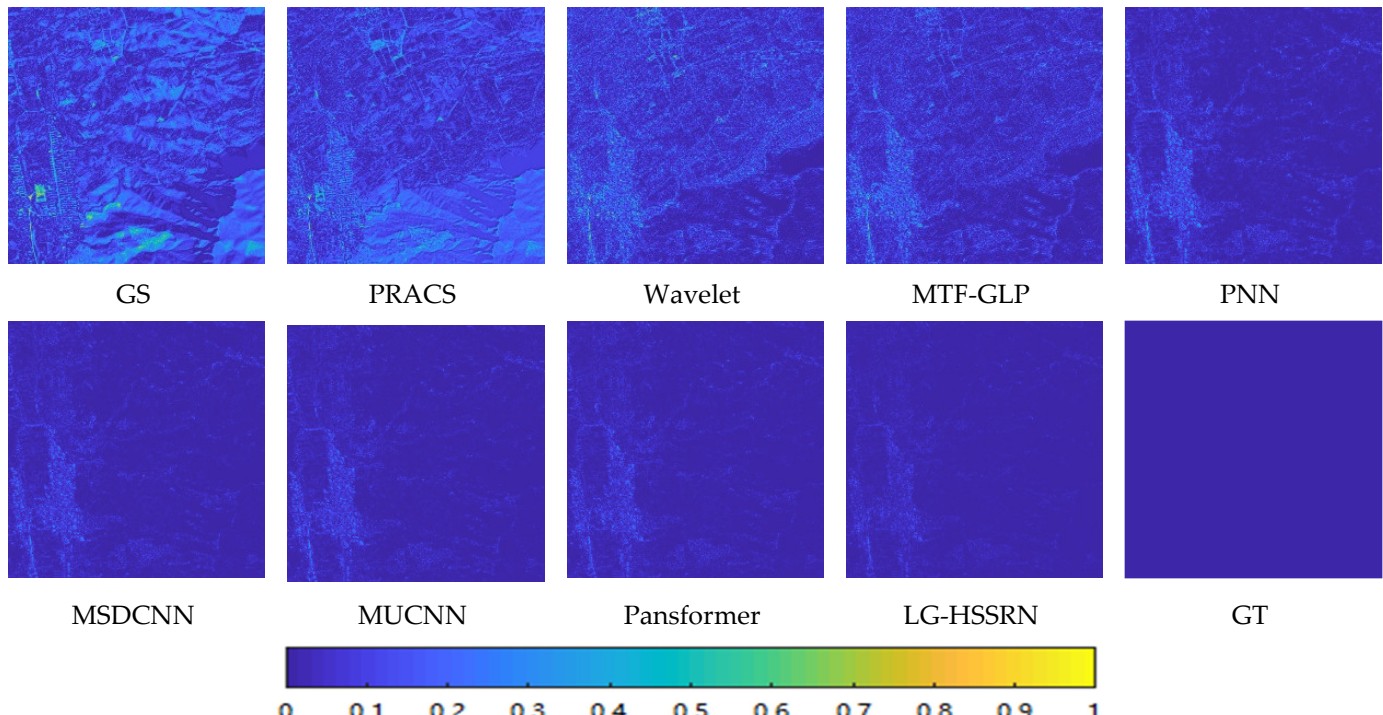

**Figure 10.** Comparison of mean square error maps of various methods on the GaoFen-2 simulated dataset.

The LG-HSSRN is closest to the ground truth image, and it is obvious that the arrangement of the houses in the local zoomed image is complete, the edges of the buildings are clear, and the color fidelity is high. The differential image in Figure 10 also confirms the superior performance of the proposed method. Table 5 highlights the performance of the proposed method in terms of index data.

**Table 5.** Quantitative evaluation metrics for the GaoFen-2 simulation dataset.

| Method | SAM | ERGAS | CC | UIQI | Q4 |
|--------|-----|-------|-----|------|-----|
| Reference | 0 | 0 | 1 | 1 | 1 |
| GS | 2.5771 | 3.3413 | 0.8922 | 0.8556 | 0.8384 |
| PRACS | 3.7160 | 3.2467 | 0.9033 | 0.8020 | 0.8018 |
| MTF-GLP | 2.2970 | 3.0897 | 0.8922 | 0.8665 | 0.8448 |
| Wavelet | 2.9142 | 3.8819 | 0.9114 | 0.8793 | 0.8582 |
| PNN | 2.1781 | 2.3802 | 0.9590 | 0.9555 | 0.9307 |
| MSDCNN | 2.1383 | 2.3538 | 0.9614 | 0.9614 | 0.9404 |
| MUCNN | 1.6530 | 2.5389 | 0.9707 | 0.9631 | 0.9432 |
| Pansformer | 1.7299 | 1.9183 | 0.9766 | 0.9739 | 0.9593 |
| LG-HSSRN | **1.2598** | **1.4595** | **0.9817** | **0.9813** | **0.9783** |

Table 6 shows the results of the evaluation metrics of all methods on the QuickBird dataset for simulation experiments, and it is seen that the LG-HSSRN is optimal in all evaluation metrics results except for the SAM spectral metrics which have a difference within 0.03 with the MUCNN.

**Table 6.** Quantitative evaluation metrics for the QuickBird simulation dataset.

| Method | SAM | ERGAS | CC | UIQI | Q4 |
|--------|-----|-------|-----|------|-----|
| Reference | 0 | 0 | 0 | 1 | 1 |
| GS | 4.3427 | 3.0762 | 0.9058 | 0.7836 | 0.7718 |
| PRACS | 3.1412 | 2.2663 | 0.9336 | 0.8433 | 0.8229 |
| MTF-GLP | 5.5898 | 3.7469 | 0.8607 | 0.7839 | 0.7639 |
| Wavelet | 3.0123 | 2.0967 | 0.9538 | 0.8437 | 0.8238 |
| PNN | 2.1483 | 1.9285 | 0.9754 | 0.9252 | 0.9034 |
| MSDCNN | 1.6763 | 1.1271 | 0.9755 | 0.9373 | 0.9221 |
| MUCNN | **1.0477** | 0.7888 | 0.9819 | 0.9591 | 0.9305 |
| Pansformer | 1.5306 | 1.0487 | 0.9822 | 0.9670 | 0.9414 |
| LG-HSSRN | 1.0759 | **0.7026** | **0.9834** | **0.9679** | **0.9487** |

Figures 11 and 12 show the visualization results and residual plots, respectively, of the corresponding methods on the QuickBird dataset. Both CS- and MRA-based methods show significant spatial and spectral distortions, and the residual plots of these methods also exhibit significant errors with the GT images. From the PNN, MSDCNN, and MUCNN fusion results, it is seen that the colors of the red houses are too light compared to the GT images and show some degree of spectral distortion. In addition, some artifacts can be observed in the results of the Pansformer method. In contrast, the LG-HSSRN is closest to the GT image, and it can also be observed that there are no obvious bright spots in the error map, which means that our method has the best fusion effect.

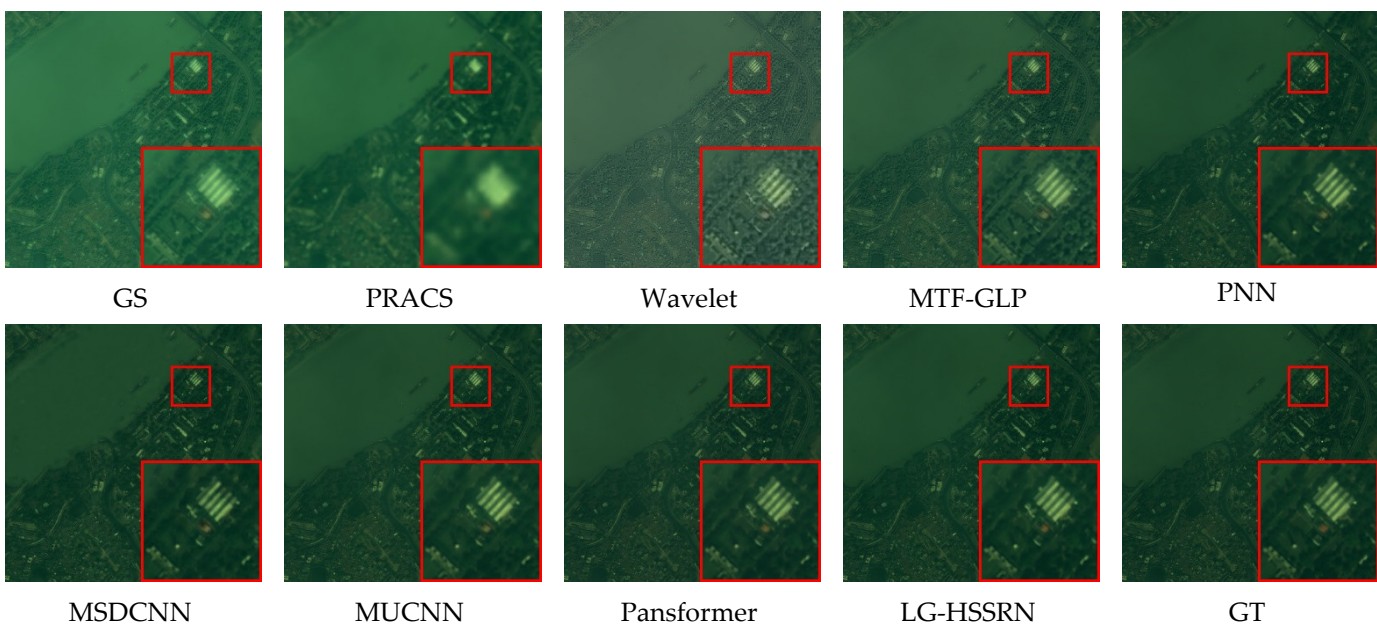

**Figure 11.** Comparison of pseudo-color maps of various methods on the QuickBird simulated dataset.

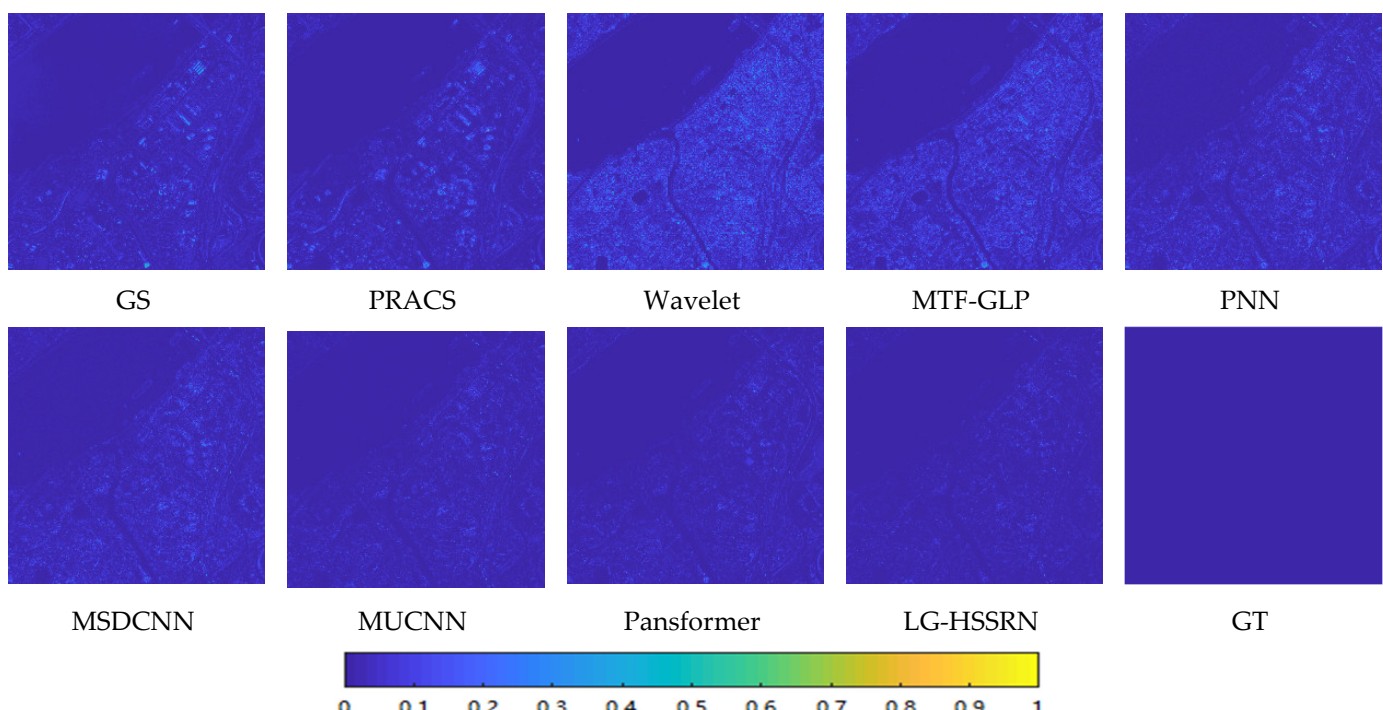

**Figure 12.** Comparison of mean square error maps of various methods on the QuickBird simulated dataset.

*3.5. Real Experiment Results and Analysis*

Figures 13–15 show the data results of real experiments with different methods on the WorldView-2, GaoFen-2 and QuickBird datasets, respectively. Figure 14 shows that the GS method has obvious spectral distortion and the color distribution of the whole image is chaotic and accompanied by spatial blurring. In contrast, the PRACS, Wavelet, and MTF-GLP methods show significant spatial distortion. For example, the PRACS and Wavelet images in Figure 13 show white artifacts, and the generated image of MTF-GLP in Figure 14 shows spatial distortion as a whole, and the outlines of roads and houses in the local detail enlargement are unclear.

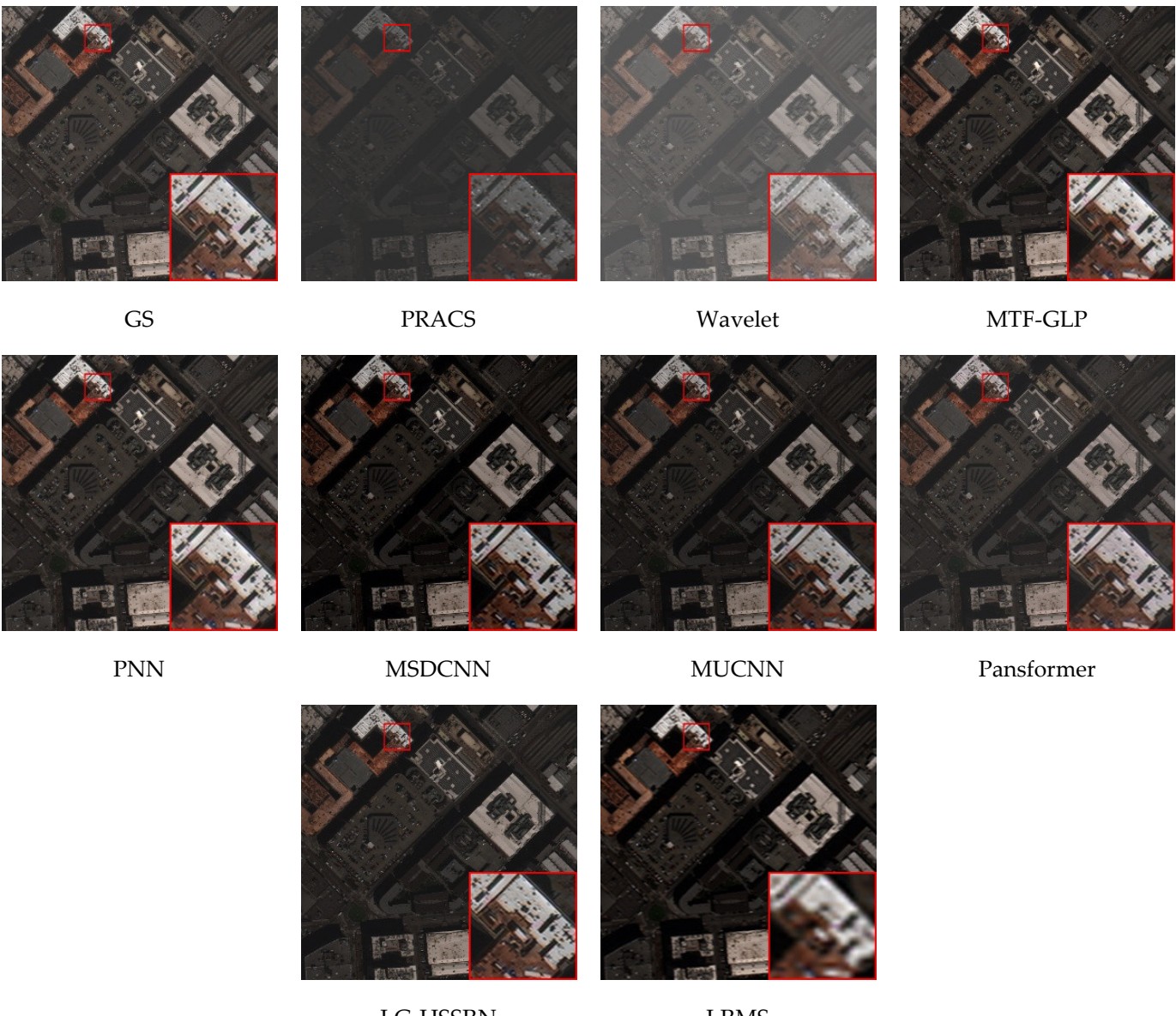

**Figure 13.** Comparison of pseudo-color maps of various methods on the WorldView-2 real dataset.

The overall spatial architecture of the images of the PNN, MSDCNN and MUCNN methods is still clear, but some detailed information is still not enhanced and complete, such as the mound in Figure 14. The fusion results of the Pansformer method are better overall, and the enhancement of some edge details is more accurate. However, compared with the LG-HSSRN, there remain some spectral distortions in its resolution-enhanced images, for example, the color degree of the red house in Figure 14 is weaker than the color degree of our generated results. In addition, the enhancement of some spatial details is not as good as the LG-HSSRN, as can be seen from the crescent-shaped white block between the white square container and the brown building in Figure 15, which is shown more completely by our method. In conclusion, the LG-HSSRN is able to improve the resolution of MS images while fully maintaining spatial and spectral information.

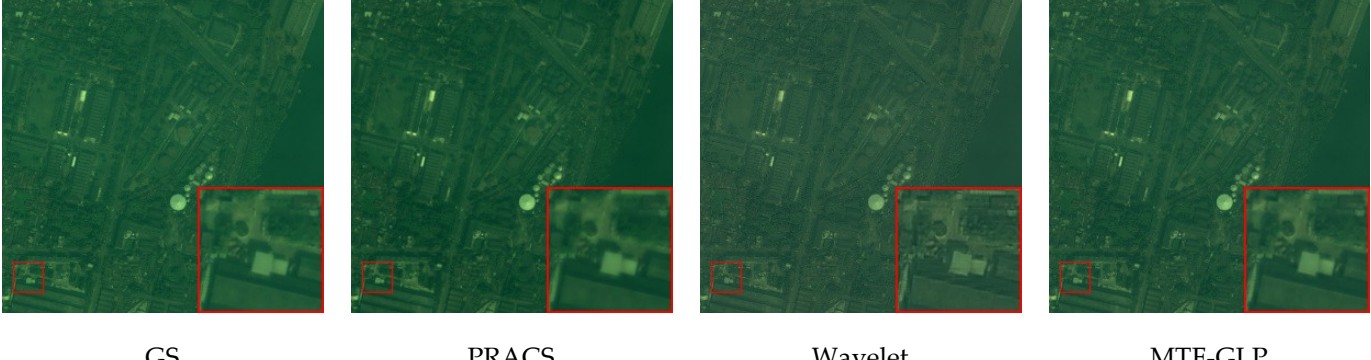

**Figure 14.** Comparison of pseudo-color maps of various methods on the GaoFen-2 real dataset.

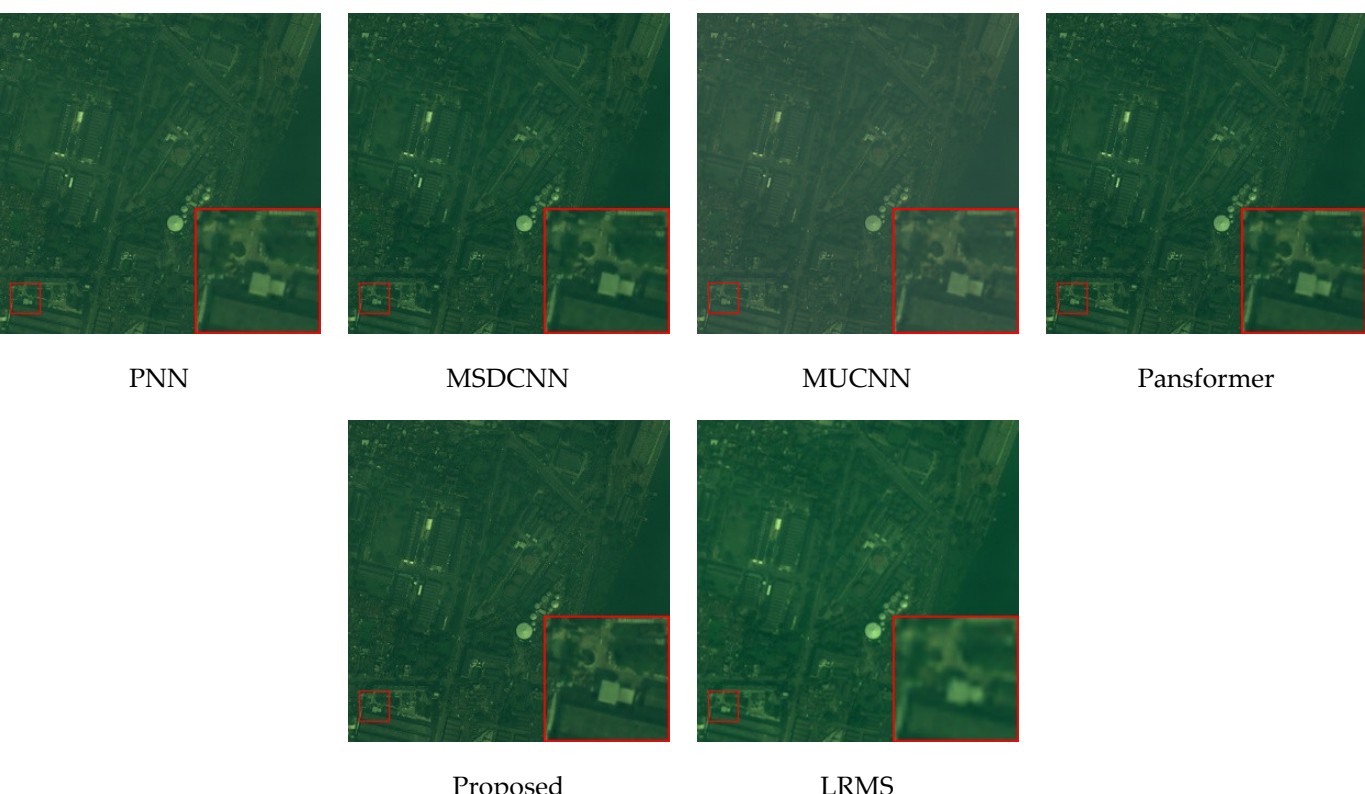

**Figure 15.** Comparison of pseudo-color maps of various methods on the QuickBird real dataset.

Table 7 shows the results of the full-resolution no-reference metrics on the three datasets, and our proposed method has the best performance in terms of overall metrics. The MUCNN and Pansformer are the next best. Compared with the first three methods, the PNN and MSDCNN fusions show spatial and spectral distortions, and both methods have higher values of the spectral metric $D_\lambda$ and spatial metric $D_s$, implying that significant spatial and spectral distortions occur. The overall indexes of both CS and MRA-based methods are lower and the fusion effect is poor. In general, the LG-HSSRN can effectively reduce the spatial and spectral distortions and retain the structural and spectral information as much as possible.

**Table 7.** Quantitative evaluation metrics for the different real datasets.

| | WorldView-2 | | | GaoFen-2 | | | QuickBird | | |
|---|---|---|---|---|---|---|---|---|---|
| | QNR | $D_\lambda$ | $D_s$ | QNR | $D_\lambda$ | $D_s$ | QNR | $D_\lambda$ | $D_s$ |
| Reference | 1 | 0 | 0 | 1 | 0 | 0 | 1 | 0 | 0 |
| GS | 0.7875 | 0.0765 | 0.1471 | 0.8387 | 0.0254 | 0.1393 | 0.7973 | 0.0279 | 0.1797 |
| PRACS | 0.8577 | 0.0497 | 0.0973 | 0.8298 | 0.0596 | 0.1175 | 0.8073 | 0.0592 | 0.1417 |
| Wavelet | 0.8670 | 0.0805 | 0.0569 | 0.8226 | 0.0806 | 0.1051 | 0.7917 | 0.1616 | 0.0556 |
| MTF-GLP | 0.8094 | 0.0176 | 0.1759 | 0.7865 | 0.0234 | 0.1945 | 0.8133 | 0.0647 | 0.1303 |
| PNN | 0.9089 | 0.0215 | 0.0710 | 0.8794 | 0.3540 | 0.0882 | 0.8923 | 0.0366 | 0.0737 |
| MSDCNN | 0.9206 | 0.0326 | 0.0482 | 0.8916 | 0.0627 | 0.0486 | 0.9135 | 0.0381 | 0.0502 |
| MUCNN | 0.9520 | 0.0304 | 0.0179 | 0.9327 | 0.0162 | 0.0518 | 0.9292 | 0.0327 | 0.0392 |
| Pansformer | 0.9545 | 0.0304 | 0.0154 | 0.9455 | 0.0182 | 0.0369 | 0.9351 | 0.0261 | 0.0396 |
| LG-HSSRN | **0.9750** | **0.0110** | **0.0140** | **0.9592** | **0.0153** | **0.0258** | **0.9479** | **0.0145** | **0.0380** |

*3.6. Performance Verification of Network Modules*

Since the texture-transformer (TT), Multi-Dconv transformer (MDT), and MSCA modules are the main contributions we present, we focus on performance validation of these modules. Table 8 shows the data results of our module validation on the WorldView-2 dataset. Figure 16 shows its visual comparison results.

**Table 8.** Quantitative evaluation metrics of different modules in the WorldView-2 dataset.

|  |  | TT | MDT | MSCG | SAM | ERGAS | CC | UIQI | Q4 |
|---|---|---|---|---|---|---|---|---|---|
| (1) | W/O(TT) |  | ✓ | ✓ | 1.8094 | 3.3359 | 0.9742 | 0.9624 | 0.9532 |
| (2) | W/O(MDT) | ✓ |  | ✓ | 3.5611 | 3.4647 | 0.9860 | 0.9526 | 0.9467 |
| (3) | W/O(MSCG) | ✓ | ✓ |  | 2.9203 | 4.1828 | 0.9728 | 0.9517 | 0.9426 |
| LG-HSSRN | All | ✓ | ✓ | ✓ | **1.5060** | **2.9222** | **0.9882** | **0.9731** | **0.9645** |

In Experiment 1 we removed the texture-transformer module, and in Experiment 2 we removed the Multi-Dconv transformer module. Removing the texture-transformer module slightly decreases the metric, but the overall numerical performance remains at a certain level. After removing the Multi-Dconv transformer module, it is obvious that the SAM spectral metric increases and the overall image evaluation metric *Q4* decreases significantly. This indicates that the Multi-Dconv transformer module is indispensable for the model.

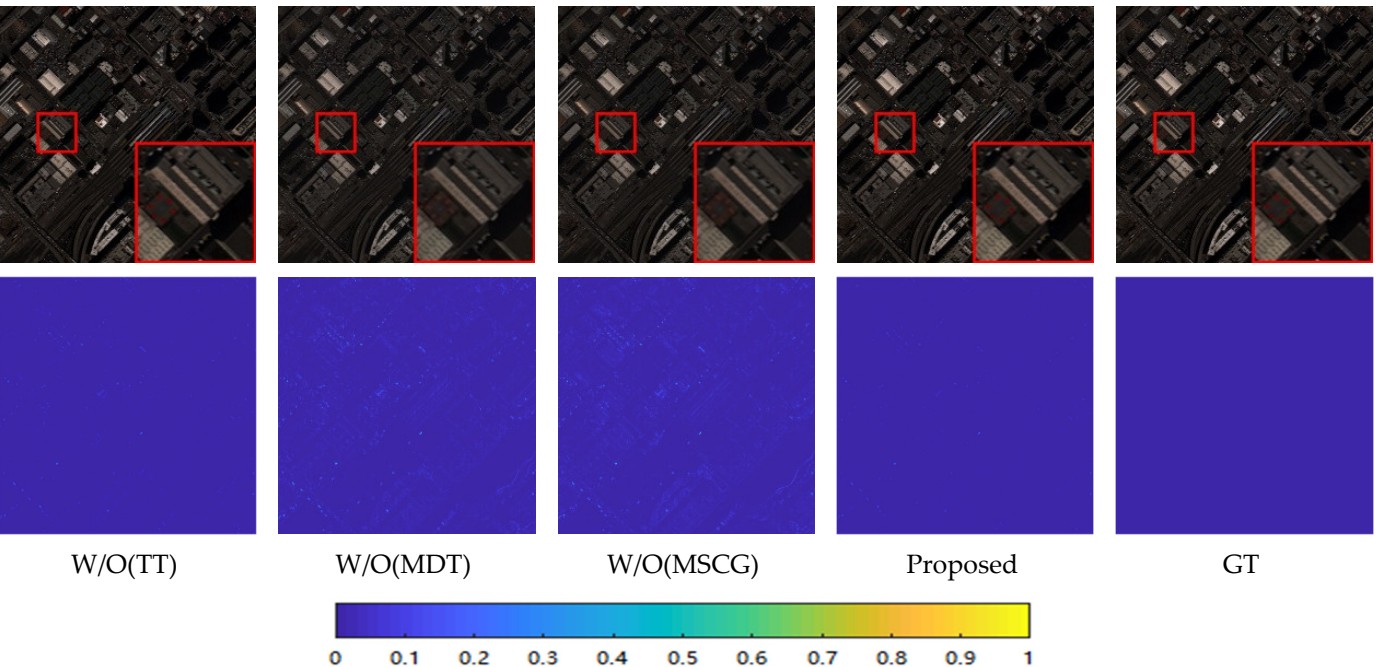

W/O(TT)        W/O(MDT)        W/O(MSCG)        Proposed        GT

0   0.1   0.2   0.3   0.4   0.5   0.6   0.7   0.8   0.9   1

**Figure 16.** Performance validation of different modules in the WorldView-2 dataset.

In contrast, the metric data from Experiment 3 with the removal of the MSCA module shows that similar high-resolution feature aggregation is essential for the pansharpening task. The MSCA module also plays a key role in the whole network.

Figure 16 shows the corresponding visual comparison plots. It can be seen that after removing the Multi-Dconv transformer and the MSCA module respectively, the red buildings in the local zoomed image appear lighter in color and the overall error increases. This also reflects the importance of several components from the side.

## 4. Discussion

Based on the analysis of the simulated and real experimental results in Sections 3.4 and 3.5 and the module performance check in Section 3.6, we give a final summary and discussion in this section. First, for the quantitative analysis and visual effect comparison of eight different comparison methods on the WorldView-2, GaoFen-2 and QuickBird datasets, we found that the GS and PRACS methods generally show spectral distortion, while their overall spatial quality is not too high and the coupling is poor for different datasets. Wavelet and MTF-GLP, two MRA-based methods, on the other hand, show significant spatial distortion, and the spatial quality of their generated results is too low, both in terms of metric data and visual perception. Among the deep learning-based methods, the PNN has different fusion effects on different datasets, poor algorithm robustness, and obvious spectral and spatial distortion, which indicates that the shallower network cannot fully acquire the deep features. The MSDCNN can fully improve the spatial quality on the basis of the PNN, and the structural information of the generated images is relatively clear, but the spectral distortion still exists. The MUCNN has good spectral retention and its spatial effect is relatively high, but the enhancement of detailed information is not sufficient, i.e., the enhancement of texture information is not sufficient. Pansformer can obtain relatively good spectral and spatial retention on some datasets. However, the overall robustness of the model is slightly lacking. The enhancement of some detailed information still needs to be improved.

The LG-HSSRN, on the other hand, can improve the overall image quality by effectively extracting global and local information. By focusing on the extraction of both channel and spatial features, our proposed model is also able to extract the corresponding features from both spatial and spectral perspectives for mass spectral images. Finally, the MSCA module is able to integrate and map the acquired features to high resolution with high representational power, which is crucial for overall image quality improvement. Simulated and real experiments on the three datasets and the final component validation experiments demonstrate the good spatial and spectral retention of our model.

In general, the proposed LG-HSSRN effectively captures spatial and channel dependencies from both global and local perspectives, achieves feature extraction from multiple sources images at multiple levels and perspectives, and fuses them by means of high-resolution representations. Compared to CNN-based pansharpening methods, our method extracts more comprehensive features; compared to other transformer-based fusion architectures, we not only capture long-range dependencies from a spatial perspective, but also extract non-local information from a channel perspective, which takes into account the spectral dimensionality of spectral images.

Of course, our proposed approach still has shortcomings in that it does not do much to deal with the redundant and discrepant information between the PAN and MS images. In future work, we will further modify the model to achieve efficient, low-redundancy image fusion. In addition, we intend to apply the model to hyperspectral and multispectral images fusion work to achieve cross-channel feature extraction of high-dimensional spectral images.

## 5. Conclusions

In this paper, we propose the LG-HSSRN for the fusion of remote sensing images. We use an LGFE module to capture local and long-range dependencies. A texture-transformer module is designed not only for learning texture features between images, but a Multi-Dconv transformer module is added for obtaining cross-channel letter context information in the global feature extraction module using the characteristics of PAN and MS images. Moreover, to better fuse the images, an MSCA module is used to obtain more representational high-resolution features. Finally, the results of simulations and real experiments on WorldView-2, GaoFen-2 and QuickBird datasets show that the LG-HSSRN exhibits the most superior performance.

**Author Contributions:** Conceptualization, W.H. and M.J.; methodology, W.H. and M.J.; software, M.J. and Z.Z.; validation, Z.Z., Q.W. and E.T.; writing—original draft preparation, W.H. and M.J.; writing—review and editing, Q.W. and E.T. All authors have read and agreed to the published version of the manuscript.

**Funding:** This research was funded by Scientific and technological key project in Henan Province, grant numbers 212102210102 and 212102210105.

**Data Availability Statement:** The data presented in this study are available in article.

**Acknowledgments:** We are very grateful to the reviewers who significantly contributed to the improvement of this paper.

**Conflicts of Interest:** The authors declare no conflict of interest.

**Abbreviation**

The abbreviations for all key terms in this article are explained below:

| | |
|---|---|
| MS | Multispectral |
| PAN | Panchromatic |
| HRMS | High-resolution multispectral |
| LRMS | Low-resolution multispectral |
| CNN | Convolutional neural network |
| LG-HSSRN | Local-global based high-resolution spatial-spectral representation network |
| LGFE | Local-global feature extraction |
| MSCA | Multi-scale context aggregation |
| MSFF | Multi-stream feature fusion |
| TT | Texture-transformer |
| MDT | Multi-Dconv Transformer |
| MSRB | Multi-scale residual block |
| Dconv | Deep convolution |
| ERGAS | The relative global synthesis error |
| SAM | Spectral angle mapper |
| CC | Correlation coefficient |
| UIOI/Q4 | Universal image quality index and its extended index |
| QNR | Reference-free quality index |

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
