# Peer review of "Local-Global Based High-Resolution Spatial-Spectral Representation Network for Pansharpening"

_remotesensing, doi:10.3390/rs14153556_

Round 1
Reviewer 1 Report
A local-global based high-resolution spatial-spectral representation network (LG-HSSRN) that captures the spatial and channel information of multispectral images from local and global perspectives is proposed in this paper. This is a very innovative work, and the effectiveness of the proposed method is also verified through experiments. I suggest some refinements are needed before publication.
1. The MUCNN method compared in the experimental part is not mentioned in the Introduction part, please introduce it.
2. At the end of the Introduction, the section introductions for this paper do not match the following section numbers, please correct them.
3. For the two data sets of WV-2,Gf-2 and QB, did the authors train only one network or train three networks separately?
4. Is the LGFE module in the three branches a module with different parameters or a consistent module? Please clarify.
5. I suggest that the reference value of the image in the table of evaluation results is placed above the index value of each comparison method to be more significant.
6. I found some typos (for example, MS image -> MS images, image block -> image patch), please correct them.
Reviewer 2 Report
This manuscript proposed a local-global based high-resolution spatial-spectral representation network (LG-HSSRN), which can effectively obtain local and global dependencies. The paper is well written and clear to follow. It would be great if you could expose your source code. Some comments:
1. In introduction part, it should give more better review on this field and add some recent literature, such as
Lu Liu, Jun Wang*, Erlei Zhang*, Bin Li, Xuan Zhu, Yongqin Zhang, Jinye Peng, “Shallow-deep convolutional network and spectral discrimination-based detail injection for multispectral imagery pan-sharpening,” IEEE Journal of Selected Topics in Applied Earth Observations and Remote Sensing, vol. 13, pp. 1772-1783, 2020.
2. In simulation experiment part, please check the tables for inconsistencies in the algorithm abbreviation and the analysis description, such as MTF-HPF, which did not appear in the previous intro.
Reviewer 3 Report
1、The paper has a vast number of abbreviations and acronyms. I suggest the authors make a table with all acronyms and abbreviations. In some cases, I had a hard time finding some of them. I believe the readers will also have this problem. 2、In lines 109,110,111, it is mentioned that “the transformer has high complexity for spatial quadratic computation on high-resolution images and ignores the channel dimensionality adaptation of the images”, but the authors don’t explain that how you solve these two problems. I suggest the authors add more details about these. 3、I believe it would be better suited if the location of the figure 3 was after it’s called in the text. This situation occurs in Figure 7, Table 3, etc.in the following text. I believe the authors will check the full text and make revisions. 4、In lines 227-230, it is mentioned that “It is worth noting that we do not perform a segmentation patch operation ...”. The authors don’t explain these in detail. And it is necessary to add how to use the multi-headed self-attentive mechanism and why it enables a good interaction of local and non-local pixels. 5、Although the meaning of (a) and (b) is marked in the legend of Figure 5, it is better that the authors need to explain in the text. Otherwise the reader will not be able to understand it in time. Some problems of the experiment: 6、The ratio of training set and test set is 4:1, why? In addition, how is the ratio of training set and validation set determined? And why? 7、Did the authors save the best model in the validation set? It should be written in the text. 8、The authors don’t show some details about hyperparameters such as learning rate, batch size, etc. I believe it will be better to show these in tabular form and explain the reason for this setting. 9、Since the authors are proposing a new method, i believe it is crucial to insert a nice and informative GitHub so that other researchers can easily replicate the experiments.I believe this work could improve by adding more details in discussion section to present the novelties and future applications more evidently. Some suggestions: (1) explain how this work advances in this field compared to other results, (2) Explain possible limitations, (3) explain possible applications. The authors could analyze this topic and explain how the proposed method could provide better results.Author Response
Please see the attachment.

Reviewer 4 Report
The authors proposed a new network architecture for pansharpening. The paper is clearly written and the proposed method itself is somewhat novel. Although the proposed method is quite complex, the contribution of each module is somewhat clear with appropriate ablation studies. Experimental results show that the proposed method outperforms the previous state-of-the-art method.
The architecture proposed in this paper could be applied to other problems. Therefore, the reviewer expects the authors to publish their implementation on Github.
